

# Controlling superfluid flows using dissipative impurities

**Martin Will**[1*], **Jamir Marino**[2], **Herwig Ott**[1] **and Michael Fleischhauer**[1]

**1** Department of Physics and Research Center OPTIMAS,
University of Kaiserslautern, 67663 Kaiserslautern, Germany
**2** Institute for Physics, Johannes Gutenberg University Mainz,
D-55099 Mainz, Germany

* willm@rhrk.uni-kl.de

## Abstract

We propose and analyze a protocol to create and control the superfluid flow in a one dimensional, weakly interacting Bose gas by noisy point contacts. Considering first a single contact in a static or moving condensate, we identify three different dynamical regimes: I. a linear response regime, where the noise induces a coherent flow in proportion to the strength of the noise, II. a Zeno regime with suppressed currents, and III. a regime of continuous soliton emission. Generalizing to two point contacts in a condensate at rest we show that noise tuning can be employed to control or stabilize the superfluid transport of particles along the segment which connects them.



# 1 Introduction

Quantum interference plays a key role in mesoscopic transport phenomena where impurities or dots are employed as 'shunts' for transferring particles, energy and information without degrading phase coherence in the process [1–3]. In recent years a novel route to investigate this field of *quantum transport* emerged by employing ultracold atoms confined by optical or magnetic potentials [4]. The ability to control and manipulate the effective dimensionality and geometry of the systems, the possibility to tune the inter-particle interaction strength, to add or eliminate disorder and to choose between fermionic or bosonic quantum particles made ultracold atoms an ideal testing ground for quantum transport phenomena [5,6]. In these systems effects are accessible which were out of reach or very challenging to investigate in solid state. E.g. the periodic velocity change of a quantum particle moving in a lattice under the action of a constant driving force, known as Bloch oscillation, is difficult to observe in condensed matter systems due to impurity scattering but has beautifully been demonstrated with ultracold atoms in optical lattices [7,8]. Transport experiments of ultra-cold Fermi atoms through point contacts [9–11] verified the quantization of conductance predicted by the Landauer theory of transport, which has previously been observed only in electronic systems. Both bosonic and fermionic superfluids can be created using ultra-cold atoms and frictionless flow has been observed [12,13]. Persistent currents in ring geometries have been realized in atomic superfluids [14,15] and cold-atom analogues of Josephson junctions have been constructed [16,17] with the potential for an atomtronic analogue of a SQUID. Finally the coupling between particle and heat transport has been observed in fermionic cold atoms providing a cold-atom analogue of the thermoelectric effect [18]. However, despite of all experimental advances in the field, the creation and precise control of superfluid currents remains a challenge in atomtronics. Besides moving potential barriers or time-dependent artificial gauge fields, currents are typically generated by a difference of chemical potentials between the ends of a channel, i.e by fixing "voltage" rather than "current".

In the present work we suggest and analyze a different method to create and manipulate the superfluid flow in a one-dimensional quasi-condensate of Bose atoms, see Fig. 1. Importantly here we control the superfluid current directly rather than fixing chemical potentials. In particular we make use of the interaction of the condensate with quantum impurities that are

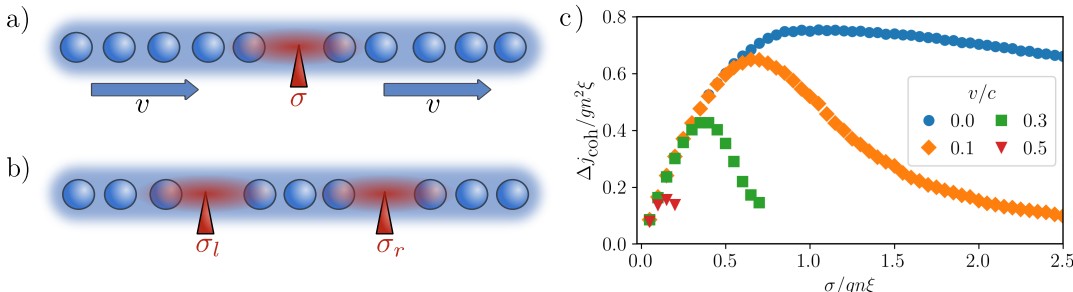

Figure 1: Scheme for controlling superfluid flow in a one dimensional interacting Bose gas using one a) or two b) noisy impurities, with or without an external current at velocity $v$. c) Induced superfluid current at a noisy point contact of noise strength $\sigma$ in a moving condensate at velocity $v$. For weak noise the current grows monotonically with $\sigma$, but for stronger noise the system enters a Zeno regime, where the current decreases. For the two largest velocities the system enters a regime of dynamical instabilities beyond a certain value of $\sigma$, where the current is no longer stationary and thus not shown.

coupled to the Boson's density with a fluctuating, i.e. noisy strength. Analyzing the system we identify different dynamical regimes, including a linear response regime, a Zeno-regime [19] with negative differential current to noise-strength characteristics, and a regime with dynamical instabilities characterized by continuous soliton emission. Impurities in interacting systems have been instrumental to develop our understanding of the extended pattern of correlations in quantum many particle systems, by employing them as probes [20–22], tunable perturbations, or even seeds for entanglement [23]. In unitary quantum dynamics, examples range from the 'catastrophic' effect of a scattering potential intruding in a Fermi sea [24, 25], to strongly entangled magnetic impurities coupled to fermionic or bosonic reservoirs, or the dressing of static and moving particles in Fermi gases or Bose-Einstein condensates as it occurs in polaron formation [26–34]. The last decade has also witnessed a growth of attention towards the dissipative counterpart of the problem of quantum impurities embedded in interacting extended quantum systems [11, 35–53]. Pioneering results of one decade ago illustrates e.g. the action of a localized dissipative potential on a macroscopic matter wave by shining an electron beam on an atomic BEC [54, 55]. Atomic losses induced by local dissipation were monitored as a function of noise strength, providing a proxy for a many-body version of the Zeno effect. The stabilisation of dark solitons by engineered losses has been studied in [56]. Fluctuations in the condensate can build up strong correlations with localized dissipation, resulting in a suppression of transport at large noise strength which can be regarded as non-equilibrium phase transition [57, 58].

In this work, we illustrate how density rearrangements provoked by local dephasing can be utilized to control coherent superflows in a one-dimensional Bose condensate. Specifically, we consider a static or uniformly moving condensate coupled to a noisy local impurity. The noisy point contact acts as a source of incoherent, i.e. non-condensed atoms, which due to total particle-number conservation creates a superfluid flow towards the impurity. The superfluid flow increases monotonically with growing noise up to some critical value at which the system becomes dominated by the quantum Zeno effect which leads to a reduction of transport corresponding to a negative differential current - noise characteristics. We furthermore demonstrate that the archetypal effect of transport suppression due to Zeno effect is drastically altered in a moving rather than a static condensate. In particular, we observe a lowering of the critical threshold of noise strength for entering the Zeno regime when the background speed of the condensate is increased. This is shown in Fig. 1c where the onset of the Zeno regime drifts towards smaller values of dissipation strength. As outreach, we demonstrate complete tunability of a supercurrent in a static condensate by a pair of noise point contacts.

## 2 Model

We consider a homogeneous one-dimensional Bose gas with weak repulsive interactions ($g > 0$) and boson mass $m$. We study the effect of a noisy point contact in a Bose gas moving relative to the impurity with fixed velocity $v$. The impurity-BEC coupling is modeled by a Gaussian white noise process $\eta(t)$, with mean $\overline{\eta(t)} = 0$ and variance $\overline{\eta(t)\eta(t')} = \delta(t - t')$, multiplied by a local potential $V(x+vt)$, whose profile will be specified subsequently. Since we consider a weakly interacting condensate [59], quantified by a small Lieb-Liniger parameter $\gamma = gm/n \ll 1$, where $n$ is the average boson density in the 1D gas, we apply a phase-space description of the quantum Bose field using the Glauber-P distribution [60]. Due to the action of the noisy point contact, we cannot ignore fluctuations even in the limit of a highly occupied condensate mode at very low temperatures. Within the phase space approach normal-ordered correlations of the Bose field operator $\hat{\psi}(x, t)$ are given by stochastic averages of a c-number field $\psi(x, t)$. The time evolution of $\psi(x, t)$ in the rest frame of the moving Bose

gas is then determined by a Gross-Pitaevskii-type equation with an additional stochastic term (SGPE) [61,62]

$$i \, \mathrm{d}\psi(x,t) = \left[ -\frac{\partial_x^2}{2m} + g|\psi(x,t)|^2 \right] \psi(x,t) \, \mathrm{d}t + V(x+vt)\, \psi(x,t) \circ \mathrm{d}W \,. \tag{1}$$

Here $dW = \eta(t)\mathrm{d}t$ is a infinitesimal Wiener process [60]. The delta-correlated white noise $\eta(t)$ results in physical systems from colored noise in the limit of small correlation times. As has been shown in [60] the SGPE Eq. (1) becomes in this limit a Stratonovich stochastic differential equation, which is denoted by the symbol "$\circ$". In order to gauge away the explicit time dependence of the potential $V(x+vt)$, we apply a Galilean transformation to the reference frame where the point contact is at rest. This results into a SGPE with static potential and with a spatial gradient term proportional to $v$:

$$i \, \mathrm{d}\phi(x,t) = \left[ -\frac{\partial_x^2}{2m} - iv\,\partial_x + g|\phi(x,t)|^2 \right] \phi(x,t) \, \mathrm{d}t + V(x)\,\phi(x,t) \circ \mathrm{d}W \,. \tag{2}$$

$\overline{\phi}(x,t)$ describes the average Bose field in the rest-frame of the impurity, which includes both a quantum mechanical average and one over classical fluctuations induced by the noisy point contact. We refer to $\overline{\phi}$ as the coherent amplitude of the Bose field.

It should be emphasised that the noise in Eqs. (1) and (2) is generated externally, e.g. by a fluctuating laser field, which is different to the SGPE derived e.g. in [63], where the noise is induced by the interaction of a thermal cloud with the condensate at finite temperature.

## 3 Noisy point contact in a static BEC

We start our analysis by reviewing the physics of a single point contact placed at $x = 0$ in a static BEC ($v = 0$). The effect of the noisy impurity on the Bose gas shares at a first sight some similarities with the physics of local losses in Bose wires [54, 55, 64, 65]: they both scatter particles out of the macroscopically populated ground state $\overline{\phi}(x,t)$. However, the dissipative impurity considered here conserves the total number of particles, which is crucial for potential applications in atomtronic devices. In order to compare with the dynamics resulting from local losses, we first analyse the coherent amplitude $\overline{\phi}$. Therefore we consider the noise average of the SGPE Eq. (2)

$$i\frac{\mathrm{d}}{\mathrm{d}t}\overline{\phi} = -\frac{\partial_x^2}{2m}\overline{\phi} + g\,\overline{|\phi|^2\phi} - \frac{i}{2}V(x)^2\,\overline{\phi} \,. \tag{3}$$

While the fluctuating potential vanishes on average it does have an effect on the average field $\overline{\phi}$. This is because it is a *multiplicative* noise and the field $\phi(t)$ at a given time depends on the noise such that $\overline{\phi(x,t)dW} \neq 0$ (Stratonovich calculus [60]). As a result of this, the average field experiences an effective loss, which physically describes nothing else than the scattering of particles out of the condensate into excited modes of the Bose gas, for more details see Appendix A.

Eq. (3) matches the evolution of the noise-averaged amplitude subject to local *particle loss* (cf. [54,55,64,65]) with the identification $V(x)^2 = 2\sigma\delta(x)$. We consider this potential as the limit of a Gaussian potential $V_l(x)^2 = 2\sigma/\sqrt{\pi l^2}\,\exp(-x^2/l^2)$, with the length $l$ acting as a regulator, such that $V(x)$ itself is well defined. If $l$ is chosen smaller than the healing length of the Bose gas $\xi = 1/\sqrt{2gnm} \gg l$ the internal structure of the impurity potential becomes irrelevant.

As shown in [54, 55, 64, 65] the effective local loss in Eq. (3) will induce currents. This can be seen most easily from the continuity equation of the modulus of the average field $|\bar{\phi}|^2$, which contains the coherent current

$$j_{\text{coh}} = \frac{1}{m} \text{Im}(\bar{\phi}^* \partial_x \bar{\phi}). \tag{4}$$

Note that here the noise is averaged over the individual fields first and then bilinear combinations are formed. $j_{\text{coh}}$ is in general not equal to the average total particle current, which is defined by deriving the continuity equation for $\phi^*\phi$ from the original SGPE, Eq. (2), and performing the noise average afterwards. The total current reads

$$j_{\text{tot}} = \frac{1}{m} \overline{\text{Im}(\phi^* \partial_x \phi)}. \tag{5}$$

We analyze both currents as well as their difference, which we refer to as the incoherent current. It describes the flow of particles in excited modes of the Bose field created by the local noise. To evaluate analytically the dynamics of Eq. (3) we assume that the nonlinear term factorizes under average $\overline{|\phi(x,t)|^2\phi(x,t)} \simeq \overline{|\phi(x,t)|^2}\,\overline{\phi(x,t)}$; this approximation turns out to be in excellent agreement with numerics provided the coherent state $\overline{\phi(x,t)}$ describing the mean-field dynamics of the Bose gas is macroscopically populated. We show the adequacy of this approximation by solving the full SGPE Eq. (2) and evaluating the coherent $|\overline{\phi(x,t)}|^2$ and total density $\overline{|\phi(x,t)|^2}$ (cf. with Fig. 2).

For weak dissipation the system is in a linear-response phase and the analytic solution of Eq. (3) reads

$$\overline{\phi(x,t)} = \sqrt{n_0}\,\exp(-im\sigma|x| - i\mu t), \tag{6}$$

(cf. also [64]). After switching on the local noise the system will assume this quasi-stationary state within a spatial region which grows in time with the local speed of sound $c_0 = \sqrt{gn_0/m}$. The density of the condensate in this area is reduced to $n_0 < n$ and the constant phase gradient describes a coherent current

$$j_{\text{coh}} = -n_0\sigma\,\text{sgn}(x), \qquad \text{for} \quad \sigma < \sigma_c, \tag{7}$$

flowing towards the point contact. As $j_{\text{coh}}$ is proportional to the noise strength $\sigma$, the regime is called "linear-response regime". Here the chemical potential is $\mu = gn_0 + m\sigma^2/2$.

Above a critical noise strength [64]

$$\sigma_c = \frac{2}{3}c = \frac{2}{3}\sqrt{2}gn\xi, \tag{8}$$

the system crosses over into a Zeno phase [19], where the current ceases to further increase with the strength of the dissipation. The critical point is reached when the velocity of the coherent flow $u_0 = j_{\text{coh}}/n_0$ attains the local speed of sound. For $\sigma > \sigma_c$ a grey soliton (a local density depletion of the size of the healing length [59]) forms at the position of the point contact, cf. with Fig. 2b. The density reduction associated with the formation of the grey soliton decreases the scattering rate at the point contact, which results in a reduction of the coherent current, which in turn determines self-consistently the depth of the grey soliton. As a consequence the functional dependence of the coherent current from the noise strenth changes from a linear increase $\sigma$ to an inverse scaling:

$$j_{\text{coh}} = -n_0 \frac{c_0^2}{\sigma}\,\text{sgn}(x), \qquad \text{for} \quad \sigma > \sigma_c. \tag{9}$$

This is characteristic of the Zeno phase in extended systems [19]: at strong enough dissipation transport across the dissipative impurity is impeded as a result of the frequent measurement of

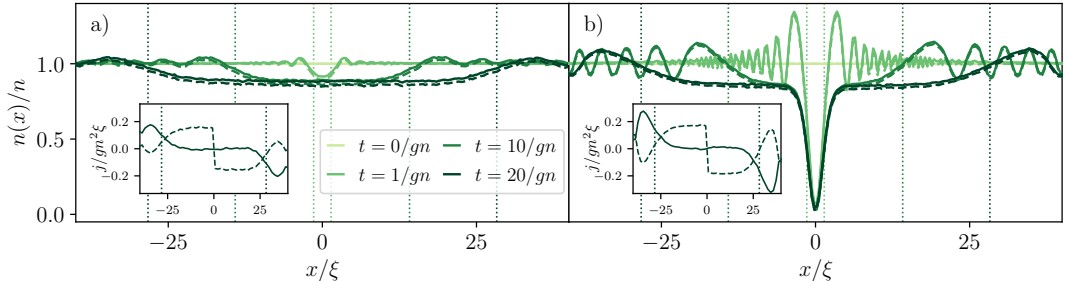

Figure 2: **Snapshots of density close to a static noisy point contact** in a) the linear-response phase $\sigma = 0.2gn\xi$ and b) in the Zeno phase $\sigma = 8gn\xi$. Solid lines are total densities $\overline{|\phi(x,t)|^2}$ while dashed lines are the coherent ones $|\overline{\phi(x,t)}|^2$. The density of incoherent particles is low. The insets show the coherent (dashed) and total (solid) particle current. The point contact scatters particles out of the coherent state, leading to a coherent current towards the noise contact and an incoherent counterflow, which exactly balances the coherent flow. The dotted vertical lines mark the size of the 'sound' cone, moving at the average speed of sound $c = \sqrt{gn/m}$. Colors match the ones of the related density profiles evaluated at the same time.

the observable to which the noise couples at $x = 0$. Outside the depleted area (which travels at the sound speed) the density $n_0 < n$ remains constant.

In contrast to the case of local loss, the noisy potential conserves the total particle number, which at first glance seems at odds with a current of particles flowing towards the point contact, while the particle density remains constant over time. Simulating the dynamics of the total SGPE Eq. (2) one finds, however, that the total current vanishes in the area of constant density, see insets of Fig. 2. This shows that the noisy point contact scatters particles out of the condensate state, causing a coherent inward flow. At the same time it is a source of particles in excited modes of the Bose field leading to an incoherent outward flow of particles. Since these particles are removed from the coherent state, the noise affects the coherent amplitude of the Bose gas similarly to a local loss. This means that a local non-unitary rearrangement of the system generates a coherent superfluid flow. In the next Sections, we harness this mechanism to engineer the coherent transport properties of the Bose gas by using purely incoherent point contacts.

## 4 Noisy point contact in a moving BEC

In this section we generalize our results to a noisy point contact in an externally imposed coherent current or equivalently to a noisy impurity moving at a constant velocity $v$ relative to the condensate. An important difference with respect to a static point contact is the emergence of a third dynamical phase (which we label 'phase III' in the following), that is characterized by the absence of stationary particle flows at the point contact. This phase occurs in addition to the linear-response I and Zeno phases II. We characterize the phases by evaluating the coherent current on the left and on the right of the noisy point contact; their difference is equal to the change in the number of particles of the coherent fraction of the field and therefore proportional to the scattering rate of particles off the dissipative impurity, cf. Appendix B Since we work in the frame in which the point contact is stationary there appears an additional term

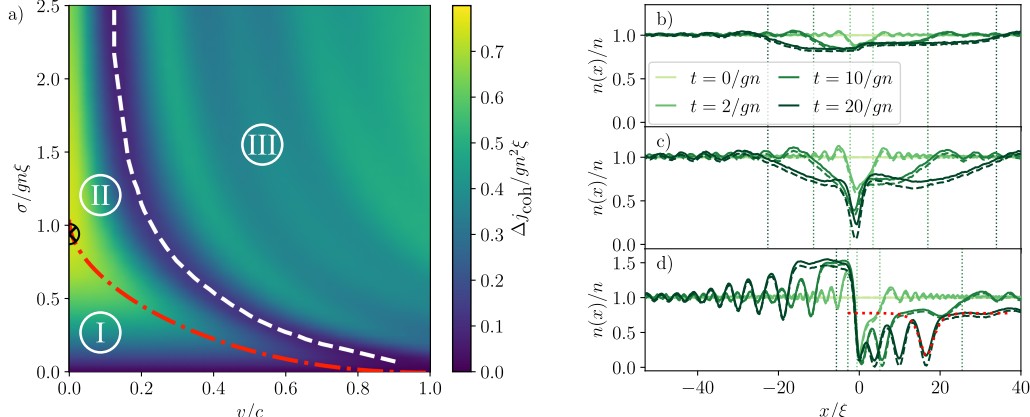

Figure 3: a) **Phase diagram of a noisy point contact** of strength $\sigma$ in a BEC moving with velocity $v$. We plot the total coherent current flowing towards the impurity on the left and on the right of the point contact, which equals the scattering rate of the dissipative impurity. We average in space over the interval $x \in \pm[2,5]\xi$ and in time over $t \in [10,20]/gn$, where the intervals are chosen to be within the 'sound' cone, but large enough to average over multiple oscillations in the dynamically unstable phase III. The red line marks the estimated transition between linear response and Zeno regime which agrees well with the local maximum for fixed $v$, see Appendix C for details. The white dashed line marks the transition form the Zeno to the soliton regime. The analytical result $\sigma_c = 2c/3$ of the transition form normal to Zeno phase at $v = 0$ [64] is shown by the black circle. b)-d) Density close to the noisy point contact at different times. Shown are in b) the normal phase I for $\sigma = 0.2gn\xi$ and $v = 0.2c$, in c) the Zeno phase II for $\sigma = 0.7gn\xi$ and $v = 0.2c$; and in d) the soliton phase III for $\sigma = 2gn\xi$ and $v = 0.8c$. The times marked in color are the same in all three plots. Solid, dashed and dotted lines are chosen as in Fig. 3. The red dotted line in d) is the profile of a grey soliton fitted to the simulated density.

in the expression of $j_{\text{coh}}$

$$j_{\text{coh}} = \frac{1}{m} \text{Im}(\bar{\phi}^* \partial_x \bar{\phi}) + v \, \bar{\phi}^* \bar{\phi} \,. \tag{10}$$

The scattering rate of particles out of the condensate and thus the total coherent current flowing towards the impurity depends both on the velocity $v$ of the point contact and on the noise strength $\sigma$ as shown in Fig. 3a. We distinguish the three phases, depending on whether this current increases or decreases as the noise strength changes.

## 4.1 Linear-response regime

Phase I, cf. with Fig. 3b, is akin to the linear-response phase of a static noisy point contact, since the scattering rate increases with increasing dissipation strength, inducing an increasing coherent current flowing towards the noise source. Due to the motion of the condensate relative to the contact, the coherent currents on the left and on the right side are unequal in magnitude. This results also in a different density on the left $n_l$ and on the right $n_r$ of the point contact. Our numerical simulations show that, like in the static case, a quasi-stationary state is established in an area growing over time, but with different velocities $(c_l - v)$ for $x < 0$ and $(c_r + v)$ for $x > 0$. The two halves of the system are characterized by different velocities for two distinct reasons: the speed of sound $c_{l,r} = \sqrt{gn_{l,r}/m}$ is different as a result of density

differences on the two sides of the dissipative impurity, and the velocity $v$ breaks the directional symmetry in the 1D gas.

## 4.2 Zeno regime

Upon increasing the dissipation strength, the system undergoes a transition into the Zeno phase II (Fig. 3c), however this occurs at a smaller critical value as in the static case. An estimate for the crossover point can be obtained as follows: The transition to the Zeno regime occurs when the local speed of sound $c(x)$ and the velocity of the coherent current $u$ become equal

$$c(x) \equiv \sqrt{\frac{g n(x)}{m}} = u(x) \equiv \frac{j_{\text{coh}}(x)}{n(x)}, \tag{11}$$

at any point in the system. The reduction of the critical noise strength in a moving condensate can then be traced back to two effects. First the coherent current is modified by to the background flow at velocity $v$. Second the local speed of sound is smaller on one side of the contact when compared to the stationary case, because of the reduced density. The overall coherent current in the system can therefore become supersonic already at a smaller critical dissipation strength. As explained in detail in Appendix C one can derive an approximate expression for the transition point by utilizing Eq. (11). The result is marked by the red line in Fig. 3a and agrees very well with the observed local maximum of the current. As in the static case a grey soliton forms in the Zeno phase II at the position of the point contact and the smaller density leads to a decrease of the scattering rate with increasing dissipation strength. However, due to the motion of the condensate relative to the point contact the coherent current cannot go to zero but must always stays finite, allowing for the onset of a new phase III.

## 4.3 Soliton-emission regime

The minimum density of the grey soliton close to the point contact would drop to zero for strong dissipation $\sigma \gg g n \xi$ obstructing any particle current at $x = 0$. However, the external flow forces particles to pass the noise contact, which can no longer be facilitated by a grey solition solution if $\sigma$ increases. This then leads to instabilities and a continuous train of solitons is formed moving in the direction of the external current, see Fig. 3d. The system becomes dynamically unstable, when the external current becomes so large that the condition for the self-consistent formation of a grey soliton can not be fulfilled any more. The minimum density in a stable grey soliton is related to the velocity $u$ of the total coherent current passing it by $n_{\text{min}}/n_0 = u^2/c_0^2$ [59]. A similar effect of a continuous creation of solitons also occurs in the case of a constant repulsive potential in a moving condensate [66]. It happens when the Bose gas density is locally reduced to an extent that a constant coherent current (superfluid flow) cannot be sustained anymore. To verify that the moving density oscillations are indeed soliton trains, we fit the analytic expression for a grey soliton wave function [59] to it, which agrees well with the observed density, see red dotted line in Fig. 3d.

In summary, a moving Bose gas responds to a local noisy impurity like a stationary Bose gas, resulting in a linear response I and a Zeno phase II with renormalized transition points between the phases. The key difference is the formation of a soliton phase III, which only exists in the presence of an external current, preventing the formation of a quasi stationary state close the impurity and a constant current flow. Different from the Zeno regime the "shooting" of solitons leads again to an increase in the time-averaged number of scattered particles with growing dissipation strength.

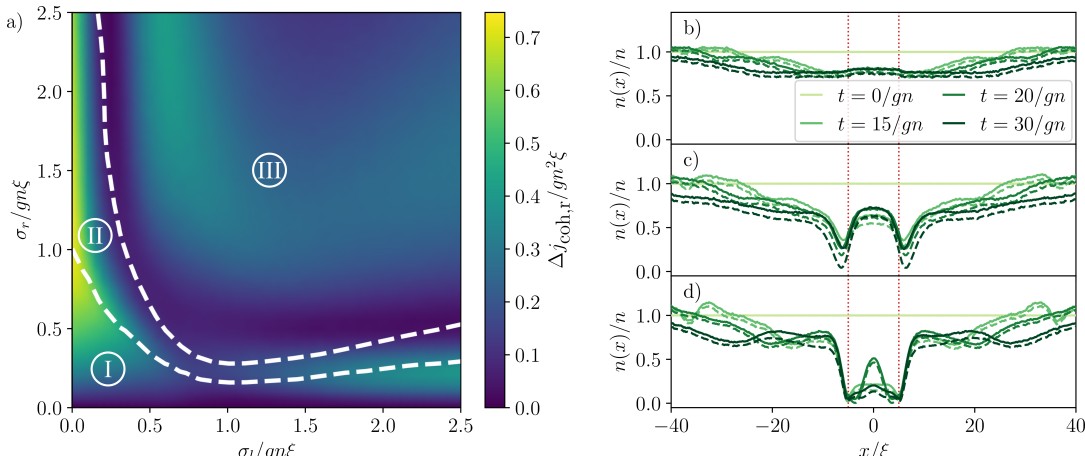

Figure 4: **Phase diagram of a configuration of two noisy point contacts.** a) Scattering rate out of the condensate at the right point contact for impurity separation $r = 10\xi$, plotted for different noise strengths $\sigma_l$ and $\sigma_r$. The current is averaged over time $t \in [25-35]/gn$ and space $x \in [-4.5, 4.5]\xi$; intervals are chosen as in Fig. 3a. The dashed lines mark the border between different phases as in Fig. 3, calculated by assuming a single contact in motion, see main text. b)-d) Density in the vicinity of two dissipative point contacts at distance $r = 10$ at equal dissipation strength $\sigma_r = \sigma_l = \sigma$. Their positions are marked with the red dotted lines. Parameters are chosen for both contacts to be in b) the normal phase $\sigma = 0.2gn\xi$, c) the Zeno phase $\sigma = .5gn\xi$ and d) the soliton phase $\sigma = 2.2gn\xi$. Solid and dashed lines are chosen as in Fig. 3.

## 5   Controlling superfluid flow with two noisy contacts

In this section we show how superfluid flow can be controlled using a pair of noisy point contacts. Each contact creates a coherent current of particles flowing towards it, which is balanced out by an incoherent one. After a time $t = r/c$, where $r$ is the distance between the contacts, the coherent current created by one reaches the other contact. Each of the two dissipative impurities thus experiences an effective coherent flow generated by the other impurity, and thus can sustain one of the three previously discussed phases. In the following we determine the phase diagram of the wire depending on the noise strength of the left ($\sigma_l$) and right ($\sigma_r$) noisy contacts. Evaluating the resulting currents in between the contacts we will demonstrate that a segment with two noisy defects at its edges, can act as a current shunt.
We assume the noises $\mathrm{d}W_r$ and $\mathrm{d}W_l$ acting on the left and right impurity to be uncorrelated $\overline{\mathrm{d}W_r\,\mathrm{d}W_l} = 0$, such that the time evolution is determined by the SGPE

$$\mathrm{d}\phi(x,t) = -i\Big[-\frac{\partial_x^2}{2m} + g|\phi(x,t)|^2\Big]\phi(x,t)\,\mathrm{d}t - i\sqrt{2\sigma_l\,\delta(x+r/2)}\,\phi(x,t) \circ \mathrm{d}W_l$$
$$-i\sqrt{2\sigma_r\,\delta(x-r/2)}\,\phi(x,t) \circ \mathrm{d}W_r. \tag{12}$$

We consider, in the following, a separation of the contacts larger than the healing length $r \gg \xi$; the latter is, in fact, the minimum length over which a coherent current can be established [59], and therefore a necessary requirement to apply the tools developed in the previous Sections. The scattering out of the condensate at the right noisy contact is plotted in Fig. 4a for different noise strengths. For a fixed noise strength of the left contact ($\sigma_l$), the number of scattered particles at the right impurity grows upon increasing the noise strength $\sigma_r$ in the linear-response phase Fig. 4b, and then it shows Zeno physics above a critical value of $\sigma_r$, see Fig. 4c. Upon

further increasing the noise strength on the right point contact, the effect of solitons 'shooting' discussed in the previous Section sets in, leading again to an increase of scattered particles when averaged over time; solitons move downstream towards the other point contact resulting in an oscillatory density patter in space and time between them (cf. Fig. 4d). We now show that the critical thresholds for the dissipation strength of two contacts can be approximated using the results for a single moving defect. Let us assume that the left contact is placed into a initially static gas. This then leads to an onset of a coherent current, as discussed in Sec. 3. We determine the velocity of this current by interpolating the results in Fig. 3a at zero velocity ($v = 0$). The right contact is then placed into this background current; we further assume that its presence will not affect the scattering rate at the left impurity and that the system near the right impurity is determined by its own dissipation strength $\sigma_r$ and the velocity of the coherent background current. Under these assumption we can determine the system response following the lines of Sec 4, and estimate the crossovers in the setup of a pair of noise contacts. These crossover points are marked with the white dashed lines in Fig. 4a and we recognize that they agree well with the observed extrema especially for small values of $\sigma_l/gn\xi$. This shows that only the coherent current is relevant for characterizing the steady state of the system under the noisy drive of the two impurities. For larger values of $\sigma_l$ the assumption of a constant coherent background current created by the left impurity no longer holds and the scattering rate out of the condensate at the left contact depends also on the noise strength of the right one. This explains the poorer agreement of the numerical results with the above physical picture for larger values of $\sigma_l$.

A possible application of the system of two noise contacts is the creation of a coherent current in the space between them. We note that in the proposed scheme the current is controlled directly and not via differences in chemical potentials. In Fig. 5 the coherent current between the contacts, averaged over space and a finite time interval, is plotted as function of the two noise strengths. Note that it is anti-symmetric since the exchange of the two interaction strengths leads to a reversal of the current. For a small sum $\sigma_+ = \sigma_r + \sigma_l \ll gn\xi$ both contacts are in the normal phase and the scattering rate of each contact is independent from the noise strength of the other. The coherent current in between the contacts is therefore the sum of two independent contacts. This is shown in the inset of Fig. 5, were the coherent current is normalized to $gn_0^2\xi_0$, with $n_0$ being the average density between the contacts, depending weakly on $\sigma_{l,r}$, and $\xi_0 = 1/\sqrt{2gmn_0}$ is the corresponding healing length. For small $\sigma_+ \equiv \sigma_l + \sigma_r$ the normalized current depends only on the difference $\sigma_- = \sigma_r - \sigma_l$ and it is equal to the current created by a single contact at dissipation strength $\sigma_-$. For larger $\sigma_+$ at least one of the two contacts is not in the linear response phase, which results in a different slope and a non monotonous dependency on $\sigma_-$.

# 6 Experimental implementation and perspectives

In this work we have revisited the Zeno crossover for particle currents traversing a moving noisy defect. We have shown that the speed of the impurity can be used as a knob to boost transport suppression. As a possible experimental implementation we envisage the use of noisy in-situ potentials, to control superfluid flows. Such potentials can be realized with two-color time-dependent optical potentials and tailored conservative potentials. We first note that it is crucial to have a vanishing mean of $V(x)$ for all positions $x$ (see section 2. This is important in order to avoid residual repulsive or attractive potentials, which interfere with the effect of the dephasing. This condition can be fulfilled by using two laser beams, which are red- and blue-detuned with respect to an atomic transition [67]. Both beams have to share the same spatial mode, which can be ensured by guiding them through the same optical fiber. For the defect

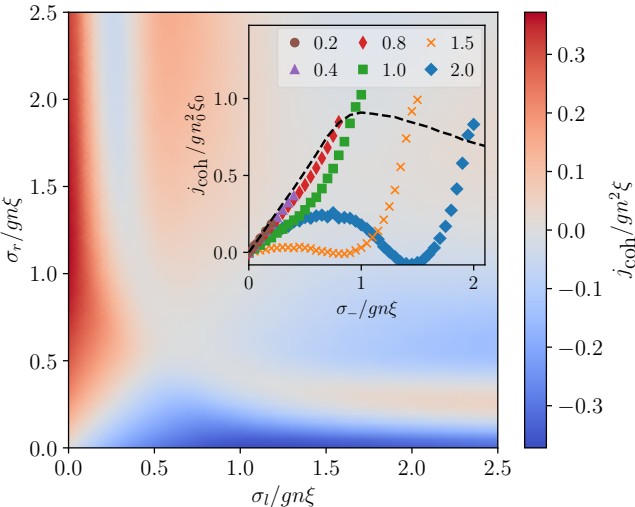

Figure 5: **Coherent current between two noisy impurities** as a function of the individual noise strengths $\sigma_r$ and $\sigma_l$. The current is averaged over time $t \in [25-35]/gn$ and space $x \in [-4.5, 4.5]\xi$. The inset shows the coherent current as a function of the difference $\sigma_-$ and sum $\sigma_+/gn\xi$ (different colors) of the two noise strengths. The current is rescaled to the density $n_0$ and the healing $\xi_0 = 1/\sqrt{2gn_0m}$ of the Bose gas inbetween the two contacts. The plot shows that the current does not depend on $\sigma_+$, for small $\sigma_+ \leq gn\xi$. The black dashed line is the coherent current created by a single stationary point contact at noise strength $\sigma_-$, which agrees well with the two-contacts result at small $\sigma_+$.

considered in this work, it is sufficient to use Gaussian beams, which are focues onto the atoms with a high numerical aperture objective. To achieve a defect size, which is smaller than the healing length (as assumed in this work), one has to find a proper combination of numerical aperture (NA=0.4 or higher is necessary for most parameter settings), a short wavelength (higher energy atomic transitions are the better choice as not much optical power is needed to create the necessary potential height) and atomic density and interaction in order to enlarge the healing length. In 1D (as considered here) or 2D configurations, the Rayleigh length should be larger than the thickness of the sample in order to treat the impurity as independent of the perpendicular direction.

Regarding the time dependence of the optical potential, a large bandwidth of the modulation is another necessity. Modulating the intensity with acousto-optical modulators typically results in a bandwidth of more than 1 MHz. This is much faster than any intrinsic timescale (interaction energy, kinetic energy, potential energy, transverse confinement) of a typical experimental setting. The corresponding correlation time of less than $1\mu s$ is therefore short enough to provide an effective $\delta$-correlated noise potential. In order to provide white or colored noise in the defect, both laser beams have to be driven with an arbitrary waveform generator, whose temporal signals are either inherently provided by the function generator or are computer generated, providing the required correlation functions. We note that experimentally, it is straightforward to generate much more complex correlation functions for the defect potential, thus bridging noisy defects and Floquet driven defects.

Measurements of the superfluid density in a quantum gas experiment are always challenging since in most schemes it is the total atomic density which is imaged. In the case of 1D systems, heterodyning with a twin system is the method of choice in order to access the motion of the superfluid as well as its amplitude [68]. To this end, one has to prepare a twin

1D system aside with the system under investigation. Upon measuring, one lets both systems interfere with each other and the fringe distance encodes the local velocity of the atoms, while the fringe contrast encodes the amplitude of the superfluid density.

From the theory side, it would be interesting to extend the control of transport properties through the segment in systems without a macroscopic condensate occupation. For instance, studying the effect of two Markovian time-dependent noisy fields coupled to local densities in an interacting fermionic wire. The non-interacting case could be solved exactly as for the single impurity [69], while the RG-scattering theory of Refs. [70] could be used to assess the role of strong quantum fluctuations in enhancing or eradicating the semi-classical effect discussed in this paper. We expect that studying real time dynamics of the problem with bosonization could serve equally well for this purpose. For what concerns the results disussed in our work, we expect that adding quantum fluctuations on top of the macroscopic occupation of the Bose gas, would not significantly alter the dynamics discussed in the paper. On one hand, quantum effects would become sizeable only on times that are parametrically large in the condensate occupation. On the other hand, the region traversed by the density waves produced by the impurity can be regarded effectively as a driven-open systems and therefore subject to decoherence: the energy is pumped into the system via the noisy contact (which is held at infinite temperature) and dissipated by the 'bath' given by the rest of the system which stays at zero temperature, till the heat front will reach it. The dynamics within the 'sound' cone will therefore wash out quantum fluctuations through decoherence as any other open quantum system would. In a semi-classical quantum trajectory description it is in fact impossible to distinguish the noise averaging used to derive the dynamics in our work, from sampling over a probability distribution function given by the quantum fluctuations inherent in the initial state: the trajectories sampled from the classical noise imprinted by the impurity would quickly dephase those arising from quantum fluctuations.

Another interesting direction would consist in generalizing the setup of our work to interacting quantum spin chains in view of applications to spintronics.

## Acknowledgements

We thank R. Barnett, J. Jager, S. Kelly and D. Sels for fruitful discussions. M.W., H.O. and M.F. acknowledge financial support by the DFG through SFB/TR 185, Project No.277625399. J. M. acknowledges financial support by the DFG through the grant HADEQUAM-MA7003/3-1. J.M. and M.F acknowledge support from the Dynamics and Topology Centre funded by the State of Rhineland Palatinate. M.W. is supported by the Max Planck Graduate Center with the Johannes Gutenberg-Universität Mainz.

## A   Derivation of the noise averaged SGPE

In the following we derive the noise average of the SGPE Eq. (12), which can be written as

$$d\phi(x,t) = A[\phi, \phi^*]\,dt + B[\phi] \circ dW, \tag{13}$$

where

$$A[\phi, \phi^*] = -i\Big[-\frac{\partial_x^2}{2m} - iv\,\partial_x + g|\phi(x,t)|^2\Big]\phi(x,t),$$
$$B[\phi] = -i\,V(x)\phi(x,t). \tag{14}$$

This equation is a Stratonovich stochastic differential equation, where the noise is correlated with $\phi(x,t)$, so that $\overline{B[\phi] \circ dW} \neq 0$. To evaluate the noise average we transform the

Stratonovich into an Ito equation, where the noise and field are not correlated $\overline{B[\phi]dW} = 0$, see [60]. The Ito equation is then given by

$$d\phi(x,t) = \left\{ A[\phi,\phi^*] + \frac{1}{2}B[\phi]\frac{\delta}{\delta\phi(x,t)}B[\phi] \right\}dt + B[\phi]dW. \tag{15}$$

The noise average results in the Gross-Pitaevskii equation for the coherent state $\overline{\phi(x,t)}$

$$\frac{\mathrm{d}}{\mathrm{d}t}\overline{\phi} = -i\left[ -\frac{\partial_x^2}{2m} - iv\,\partial_x - \frac{i}{2}V(x)^2 \right]\overline{\phi} - i\,g\,\overline{|\phi|^2\phi}. \tag{16}$$

The complex potential shows that the local noise scatters particles out of the coherent state $\overline{\phi}$, resulting in the incoherent current flowing away form the noise source.

## B Coherent particle number

In this section we show that the jump in the coherent current between the left and right sides of a noise contact is equal to the change in the number of particles of the coherent fraction of the field. We start by deriving a continuity equation for the modulus of the average field $n_{\mathrm{coh}}(x,t) = \overline{\phi}^*\,\overline{\phi}$ from the noise averaged mean-field equation Eq. (16)

$$\partial_t n_{\mathrm{coh}}(x,t) + \partial_x j_{\mathrm{coh}}(x,t) = -2\sigma\delta(x)n_{\mathrm{coh}}(x,t). \tag{17}$$

Note that since the left hand side of this equation is nonzero $N_{\mathrm{coh}} = \int_{-L/2}^{L/2} dx\, n_{\mathrm{coh}}(x,t)$ is not conserved. The local noise scatters particles out of the coherent state. This is follows from integrating Eq. (17) over the whole system

$$\dot{N}_{\mathrm{coh}} = \int_{-L/2}^{L/2} \mathrm{d}x\,\partial_t\,n_{\mathrm{coh}}(x,t) = -\sigma\,n_{\mathrm{coh}}(0,t), \tag{18}$$

where the current term vanishes because we use periodic boundary conditions. Integrating Eq. (17) again, but over a small interval around the impurity shows

$$\Delta j_{\mathrm{coh}} = \int_{-\epsilon}^{\epsilon} \mathrm{d}x\,\partial_x j_{\mathrm{coh}} = -2\sigma\,n_{\mathrm{coh}}(0,t), \tag{19}$$

where we used that $n_{\mathrm{coh}}(x,t)$ is constant close to the impurity in the long time limit, which we showed by simulating the dynamics of the total SGPE Eq. (2). This yields

$$\dot{N}_{\mathrm{coh}} = \Delta j_{\mathrm{coh}}. \tag{20}$$

## C Estimate of the transition point between linear response and Zeno regimes

In the following we estimate the linear response to Zeno transition of a noisy point contact in an external driven current of velocity $v$. We do so by deriving four equations containing the local speeds of sound $c_i = \sqrt{g n_i/m}$ and current velocities $u_i = j_{\mathrm{coh},i}/n_i$ at the left ($i = l$) and right side ($i = r$) of the noise contact, which determine the crossover point.

The system undergoes a transition, once the current velocity is equal to the speed of sound

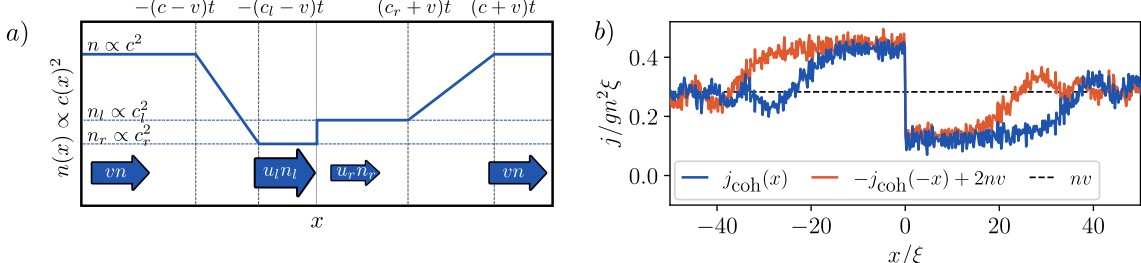

Figure 6: a) Qualitative sketch of the density profile (i.e. local speed of sound) at a dissipative point contact in a constant background current $vn$, within the linear response regime. The size of the blue arrows indicates the strength of the current at their position. b) Simulation of the coherent current in the linear response regime ($\sigma = 0.2gn\xi$, $v = 0.2c$ and $t = 20/gn$). The two lines agree close to the contact, since it induces equally strong currents on both sides.

$c_i = |u_i|$ on either of the two sides of the contact. For $v > 0$ the simulation show $c_l < c_r$ and $|u_l| > |u_r|$, see Fig. 6, causing the critical condition to be fulfilled first on the left side

$$c_l = u_l, \tag{21}$$

which is the first equation we use. We determine the other three by analyzing the system in the linear response regime and assuming that the conditions are still valid at the critical point. Since the state in the depleted area is quasi stationary, the chemical potential

$$\mu = gn_i + \frac{1}{2}mu_i^2 - \frac{1}{2}mv^2, \tag{22}$$

on both sides of the contact must be equal, from which the second equation is derived

$$c_l^2 + \frac{1}{2}u_l^2 = c_r^2 + \frac{1}{2}u_r^2. \tag{23}$$

For the third equation we utilize the numerical evidence, that the contact induces equally strong currents on both sides, which is either added to or subtracted form the background current $vn$, see Fig. 6b. This symmetry can be written as

$$j_{\mathrm{coh},r} + j_{\mathrm{coh},l} = 2vn \;\Rightarrow\; c_l^2 u_l + c_r^2 u_r = 2vc^2. \tag{24}$$

At last we derive an equation for the difference of the currents, which is equal to the change in the number of particles of the coherent fraction of the field $\dot{N}_{\mathrm{coh}}$, see Eq. (20). We assume that these particles are only removed from the "transition area" in between the quasi stationary state at density $n_i$ and the unperturbed area at density $n$. To estimate it we approximate the density profile as being linear, as illustrated in Fig. 6a. This results in

$$\dot{N}_{\mathrm{coh}} = -\frac{1}{2}(n - n_l)(c_l + c - 2v) - \frac{1}{2}(n - n_r)(c_r + c + 2v). \tag{25}$$

The fourth equation is then given by

$$2c_l^2 u_l - 2c_r^2 u_r = (c^2 - c_l^2)(c_l + c - 2v) + (c^2 - c_r^2)(c_r + c + 2v). \tag{26}$$

To determine the critical values we solve Eqs. (21), (23), (24) and (26) numerically and in order to calculate the corresponding critical noise strength $\sigma_c$ we use Eq. (19), with the approximation $n_{\mathrm{coh}}(0, t) = (n_r + n_l)/2$. This eventually yields

$$\sigma_c = \frac{c_l^2 u_l - c_r^2 u_r}{c_l^2 + c_r^2}. \tag{27}$$

The critical dissipation strength derived in this way agrees very well with the local maximum in the coherent current, which we calculated numerically, see Fig. 3a. In the stationary case ($v = 0$) we get $\sigma_c = 0.74c$, which is only slightly larger as the exact value $\sigma_c = 2c/3$ [64].

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
