# Peer review of "Controlling superfluid flows using dissipative impurities"

_SciPost Physics, doi:SciPost Phys. 14, 064 (2023)_

## Round 1 · Referee Report · Anonymous (Referee 1) · 2022-12-12

Strengths
1- interesting physics effect
Report
The authors investigate the effect of a moving impurity in a BEC. The
impurity consists in a nosy point contact. The main result of the paper is that in contrast with the case of a static impurity, where there are two regimes, namely a linear response regime and a Zeno one, a new regime emerges if the impurity is moving. Physically, this means that the current attributable to the impurity motion is so strong that a quasi stationary state around the impurity cannot form.
The results of the paper are interesting and the authors discuss the physics of the problem in a thorough way. The paper is well written.
I recommend the paper for publication in Scipost Physics.
impurity consists in a nosy point contact. The main result of the paper is that in contrast with the case of a static impurity, where there are two regimes, namely a linear response regime and a Zeno one, a new regime emerges if the impurity is moving. Physically, this means that the current attributable to the impurity motion is so strong that a quasi stationary state around the impurity cannot form.
The results of the paper are interesting and the authors discuss the physics of the problem in a thorough way. The paper is well written.
I recommend the paper for publication in Scipost Physics.

Anonymous on 2022-10-21 [id 2939]
This is very interesting research. At this point I have some questions about technical aspects of the modelling.
As a general comment, the motivation of an SGPE via the Glauber-P function, along the lines of the Stoof derivation, is, to me at least, quite unnatural. A consistent derivation of the Bose-gas reservoir interaction can be carried out in the truncated Wigner approximation, as developed in
[1] C. W. Gardiner and M. J. Davis, The Stochastic Gross–Pitaevskii Equation: II, Journal of Physics B: Atomic, Molecular and Optical Physics 36, 4731 (2003).
The theory is developed by introducing a formal energy cutoff to define the reservoir, an approach with a number of advantages. First, it is clear what is meant by the low energy "c-field" (everything below the cutoff). Second, the noise terms do not suffer any ultra-violet divergence. Third, and most interesting for the present work, there are two classes of interaction appearing in the resulting stochastic projected Gross-Pitaevskii equation (SPGPE).
Number-damping interactions in which particles are exchanged with the reservoir (incoherent region) until equilibrium is reached.
Energy-damping interactions in which only energy is exchanged with the reservoir. This interaction generates a term of potential type, and an associated (multiplicative) noise. There is a specific way that the potential depends on the state of the field (roughly speaking, the divergence of the c-field current), but the general form is similar to that presented in this work, but with an additional energy damping term. It is worth pointing out that this interaction doesn't appear in the Stoof SGPE, due to the specific choice of a very low cutoff energy (the condensate energy); this choice is linked to avoiding the issue of how to implement the cutoff in practice, a problem that is solved in [1].
Several later works have further developed the formalism, and carried out numerical simulations of the complete theory, e.g
S. J. Rooney, P. B. Blakie, and A. S. Bradley, Stochastic Projected Gross-Pitaevskii Equation, Physical Review A 86, 053634 (2012). A. S. Bradley, S. J. Rooney, and R. G. McDonald, Low-Dimensional Stochastic Projected Gross-Pitaevskii Equation, Physical Review A 92, 033631 (2015). R. G. McDonald, P. S. Barnett, F. Atayee, and A. Bradley, Dynamics of Hot Bose-Einstein Condensates: Stochastic Ehrenfest Relations for Number and Energy Damping, SciPost Physics 8, 029 (2020).
For a review, see P. B. Blakie, A. S. Bradley, M. J. Davis, R. J. Ballagh, and C. W. Gardiner, Dynamics and Statistical Mechanics of Ultra-Cold Bose Gases Using c-Field Techniques, Advances in Physics 57, 363 (2008).
So my questions are about the SGPE used in the modelling, and the nature of the point contact:
Given that the equation (1) isn't strictly the usual reservoir interaction SGPE, or SPGPE, I wonder if this is really the clearest way to motivate the use of Equation (1)?
It is clear that if the interaction is not coming from a reservoir interaction, but rather just a noisy potential, then damping would not occur. On the other hand, if the impurity is dissipative, as in the current work, then the damping and noise terms are essential and the relationship between them is important. Without damping it is not possible for the field to reach equilibrium with the impurity. What is the physical reason why there is no explicit damping term associated with the impurity noise?
Without a microscopic proof of the SGPE used to model the point contacts, the theoretical background reads as phenomenology. This is of course fine, but is there a fundamental reason why a derivation can't be carried out or is challenging?
Along similar lines, why is Stratonovich used over Ito? The motivation appears to be that the mean field equation has an associated loss term if one starts from Stratonovich. But if (1) is reinterpreted as Ito, then the mean field equation (3) would be lossless for any potential. Ito SGPE would seem a clearer choice on physical grounds for a noisy potential, namely that the potential and the field are uncorrelated in the interval dt. By choosing Stratonovich there is a correlation between the field and noise, and an effective loss term that doesn't come from a microscopic reservoir theory. So: how physical is the proposed noise?
Choosing a Stratonovich interpretation leads to the argument below Eq (3) for an effective loss describing "nothing else than the scattering of particles out of the condensate into excited modes of the Bose gas" However, if this is physically accurate it must be possible to capture such a loss using a number-damping theory, rather than the number-conserving interaction used in this work. My main question is whether Stratonovich SGPE is really the right choice for a noisy potential? It would be helpful to give some further discussion around the use of (1), and what the physical implications are. For example, in Ref. [53] the noisy defect is introduced as phenomenology, while in Ref. [54] a one-body loss term is presented as the result of a Born-Markov master equation. In the present manuscript one gets the impression that there are deeper arguments, but is not clear what they are. It would be helpful to the reader to either say it is phenomenology, or summarize the derivation, or at least the physical setup for the derivation.
In deriving (2) there is now an effective damping term in the frame of the impurity, and the system could eventually come into equilibrium in that frame. Is this to be expected physically? Or are there obvious grounds to discount such a final state due to the nature of the impurity?

---

## Round 2 · Author Response

Dear Editor, Thank you for considering our manuscript “Controlling superfluid flows using dissipative impurities”. Based on the reviewers comments, we have revised our manuscript and would like to resubmit it to SciPost Physics. Our responses to the reviewers comments can be found below. Your sincerely Martin Will, Jamir Marino, Herwig Ott, and Michael Fleischhauer

Reply to “Anonymous Report 1 on 2022-12-12 (Invited Report)”:

We like to thank the referee for the careful reading and for recommending our paper for publication in Scipost Physics.

Reply to “Anonymous on 2022-10-21 [id 2939]”:

We like to thank the referee for the careful reading and the helpful comments. In the following we will reply to them. We would like to emphasize that the SGPE Equation (1), is different from the one derived in [1] C. W. Gardiner and M. J. Davis, The Stochastic Gross–Pitaevskii Equation: II, Journal of Physics B: Atomic, Molecular and Optical Physics 36, 4731 (2003). The theory in [1] describes a Bose gas at finite temperature and the noise is caused by the interaction of the thermal reservoir with the condensed atoms. In contrast to this our work analyses the effect of a noisy potential onto a Bose gas, which is initially at zero temperature. The noise is induced externally, for example by a fluctuating laser field, see section 6 for more details. Answering the referees questions about the SGPE used in our work:

  1. -5. To derive equation (1) (for v=0) we start with a Gross-Pitaevskii equation of a Bose Gas with a time dependent potential V(x,t), as derived in [2] C. J. Pethick and H. Smith, Bose-Einstein condensation in dilute gases, Cambridge University Press, and assume that the potential fluctuates globally, i.e. its spatio-temporal dependence factorizes in time and space V(x,t) = V(x) \eta(t). The term \eta(t) models the time-dependent fluctuations, which can be implemented in experiments by a noisy laser field. Since real fluctuations always have a finite correlation time, \eta(t) is a colored noise process. In the limit where the correlation time is short compared to all other timescales of the system it can be approximated by a delta correlated white noise process. As shown in section 6.5 of Ref. [3] C. Gardiner, Handbook of stochastic methods for physics, chemistry, and the natural sciences, Springer (1985), a white noise process which is a limit of a non-white noise process results in a Stratonovich stochastic differential equation, which is the reason it is used here rather than an Ito equation. However, the Stratonovich equation can be transformed into an Ito equation which then contains an effective damping term, see Equation (14) and (15).

  2. Note that equation (2) contains no damping term, since also the term “-i v \delta_x” generates unitary dynamics. It remains an open question whether the system eventually reaches thermal equilibrium, however the present work mainly deals with the short to intermediate time evolution.

We clarified the mentioned points in the revision of the manuscript.

---

## Round 2 · List of Changes

1. A paragraph after Eq. (1) has been changed to clarify why a Stratonovich stochastic differential equation is used, rather than an Ito one.
2. Added a paragraph at the end of section 2 to emphasize that the origin of the noise in the stochastic Gross-Pitaevskii equation Eqs. (1) and (2) is different from that in previous work.

You are currently on this page

Resubmission scipost_202210_00044v2 on 14 December 2022

---

## Editorial Decision

published